# Hidden phase in a two-dimensional Sn layer stabilized by modulation hole doping

Fangfei Ming[1,*], Daniel Mulugeta[1,*], Weisong Tu[1], Tyler S. Smith[1], Paolo Vilmercati[1,2], Geunseop Lee[3], Ying-Tzu Huang[4], Renee D. Diehl[4], Paul C. Snijders[1,5] & Hanno H. Weitering[1,5]

Semiconductor surfaces and ultrathin interfaces exhibit an interesting variety of two-dimensional quantum matter phases, such as charge density waves, spin density waves and superconducting condensates. Yet, the electronic properties of these broken symmetry phases are extremely difficult to control due to the inherent difficulty of doping a strictly two-dimensional material without introducing chemical disorder. Here we successfully exploit a modulation doping scheme to uncover, in conjunction with a scanning tunnelling micro-scope tip-assist, a hidden equilibrium phase in a hole-doped bilayer of Sn on Si(111). This new phase is intrinsically phase separated into insulating domains with polar and nonpolar symmetries. Its formation involves a spontaneous symmetry breaking process that appears to be electronically driven, notwithstanding the lack of metallicity in this system. This modulation doping approach allows access to novel phases of matter, promising new avenues for exploring competing quantum matter phases on a silicon platform.

[1] Department of Physics and Astronomy, The University of Tennessee, Knoxville, Tennessee 37996, USA. [2] Joint Institute for Advanced Materials at The University of Tennessee, Knoxville, Tennessee 37996, USA. [3] Department of Physics, Inha University, Inchon 402-751, Korea. [4] Department of Physics, Penn State University, University Park, Pennsylvania 16802, USA. [5] Materials Science and Technology Division, Oak Ridge National Laboratory, Oak Ridge, Tennessee 37831, USA. * These authors contributed equally to this work. Correspondence and requests for materials should be addressed to P.C.S. (email: snijderspc@ornl.gov) or to H.H.W. (email: hanno@utk.edu).

Some of the conceptually most interesting materials in condensed matter research feature interacting quasiparticles that are associated with their charge, spin and lattice degrees of freedom. The many-body nature of these interactions often produces 'quantum matter' condensates characterized by macroscopic quantum coherence, new order-parameter symmetries and alluring physical properties. Prominent examples include charge- or spin-density wave condensates[1], magnetism[2] and superconductivity[3]. The deliberate introduction of excess charge carriers in a parent material through chemical doping has been a key strategy to access these phases and tailor their properties. A particularly appealing scheme is modulation doping, in which chemical impurities are placed in the electronically inactive regions of a solid-state heterostructure while donating their valence charge to the active regions[4]. Because the dopants and free carrier regions are spatially separated, carrier doping can be accomplished without significantly modifying the potential landscape of the material of interest. Indeed, modulation doping has been a key strategy in the search for novel quantum matter phases such as fractional quantum Hall states in semiconductor heterostructures[5] and unconventional superconductivity in layered perovskites[6].

Crystal surfaces are inherently two dimensional (2D), and there is an increasing body of evidence that simple monatomic adsorbate layers on semiconductor surfaces exhibit hallmarks of quantum matter, such as long-range quantum coherence[7,8] and spontaneous symmetry breaking[9,10]. These systems are of particular interest because electronically they are strictly 2D. Attempts to dope these surface systems via deposition of, for example, alkali metal atoms have had mixed success[11–15]. Low doses of alkali adsorbates can shift the transition temperature of a Peierls-like surface instability[14,15], but it remains difficult to attribute the temperature shift exclusively to charge doping as the adsorbate atoms introduce many defects. Higher alkali coverages amount to overall chemical modification and, hence, property tuning of the parent material is very limited. In fact, no new phases have emerged from this approach. Clearly, a modulation doping scheme for surfaces and interfaces would be highly desirable in the search for novel 2D quantum materials.

Here, we successfully control the n-type and p-type carrier density of the Si(111)$(2\sqrt{3} \times 2\sqrt{3})R30°$-Sn surface reconstruction[16] by changing the chemical potential in the bulk substrate while leaving the chemical composition and structural integrity of the surface intact. The $(2\sqrt{3} \times 2\sqrt{3})R30°$ surface reconstruction retains its translational symmetry upon electron doping, but the hole-doped reconstruction phase-separates into insulating $(2\sqrt{3} \times 2\sqrt{3})R30°$ and $(4\sqrt{3} \times 2\sqrt{3})R30°$ domains. The formation mechanism of this novel $(4\sqrt{3} \times 2\sqrt{3})R30°$ phase involves a rare displacive transition, assisted by proximity coupling to the tip of a scanning tunnelling microscope (STM). This successful demonstration of the modulation doping scheme, and particularly its ability to unveil a hitherto inaccessible hidden state of matter, may pave the way for the discovery of other emergent phases on crystal surfaces, possibly including superconductivity[17].

## Results

**Modulation doping of the surface reconstruction.** The n-type Si(111)$(2\sqrt{3} \times 2\sqrt{3})R30°$-Sn reconstruction was prepared by depositing approximately one monolayer (ML) of Sn on the $7 \times 7$ reconstructed surface of a heavily doped n-type Si(111) wafer (see Methods). Likewise, hole doping is accomplished by growing an identical $(2\sqrt{3} \times 2\sqrt{3})R30°$ Sn layer on boron-doped substrates with either a $7 \times 7$ or a B-induced $(\sqrt{3} \times \sqrt{3})R30°$-B surface reconstruction. As discussed in Supplementary Note 1,

any change in the bulk chemical will lead to a readjustment of the chemical potential in the surface layer. This readjustment involves charge transfer between the surface states and the bulk, and generally implies doping by the excess charges or 'dopant charges' in the surface states. The resulting band alignments for the n- and p-type systems are shown in Supplementary Fig. 1.

The $(2\sqrt{3} \times 2\sqrt{3})R30°$ Sn layer grown on the $(\sqrt{3} \times \sqrt{3})R30°$-B surface reconstruction corresponds to the highest hole-doping level where boron atoms have segregated towards the surface, populating the $S_5$ lattice locations below the Si adatoms[18]; see Fig. 1a. If these boron atoms are all ionized, the hole concentration would amount to $\sim 0.3$ hole per Sn atom. From here on, we label the $(2\sqrt{3} \times 2\sqrt{3})R30°$-Sn layers on the n-type and p-type $7 \times 7$ substrates as n-$2\sqrt{3}$Sn and p-$2\sqrt{3}$Sn, respectively. Those on the $(\sqrt{3} \times \sqrt{3})R30°$-B substrate are labelled B-$2\sqrt{3}$Sn.

Figure 1b shows a schematic atomic model of the B-$2\sqrt{3}$Sn structure, similar to an earlier proposed model[16] but taking into account the presence of boron at the $S_5$ lattice location. In this model, boron dopants are separated from the Sn atoms by at least

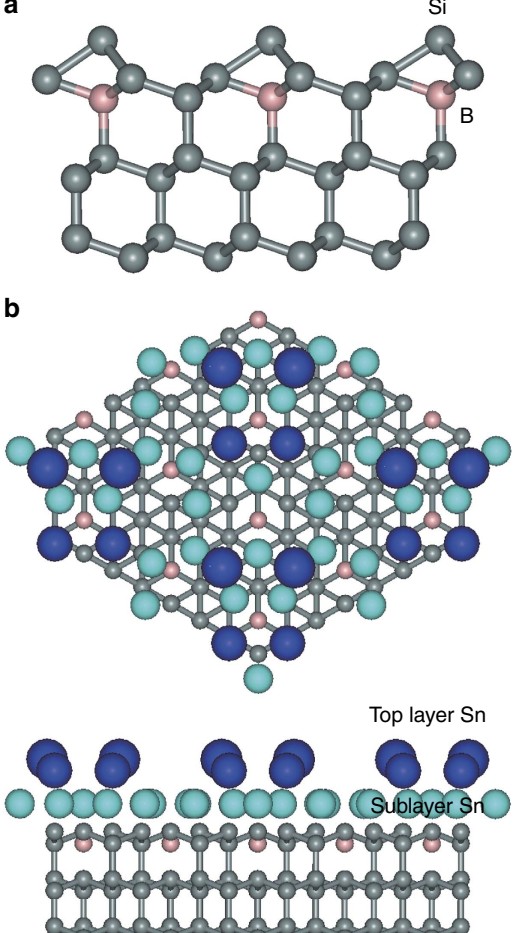

**Figure 1 | Structural model of the Sn/Si(111)$(2\sqrt{3} \times 2\sqrt{3})R30°$-B interface based on the Tornevik model.** (**a**) Side view of the Si(111)$(\sqrt{3} \times \sqrt{3})R30°$-B substrate. The subsurface boron atoms at the $S_5$ lattice location are coloured in pink. (**b**) Top view and side view of the Sn/Si(111)$(2\sqrt{3} \times 2\sqrt{3})R30°$-B interface, according to ref. 16, but with the subsurface boron atoms included (pink). The Sn tetramer units are coloured dark blue. Note that there is no consensus in the literature regarding the structure of the Sn/Si(111)$(2\sqrt{3} \times 2\sqrt{3})R30°$-B interface, and the model presented here serves only for illustrative purposes.

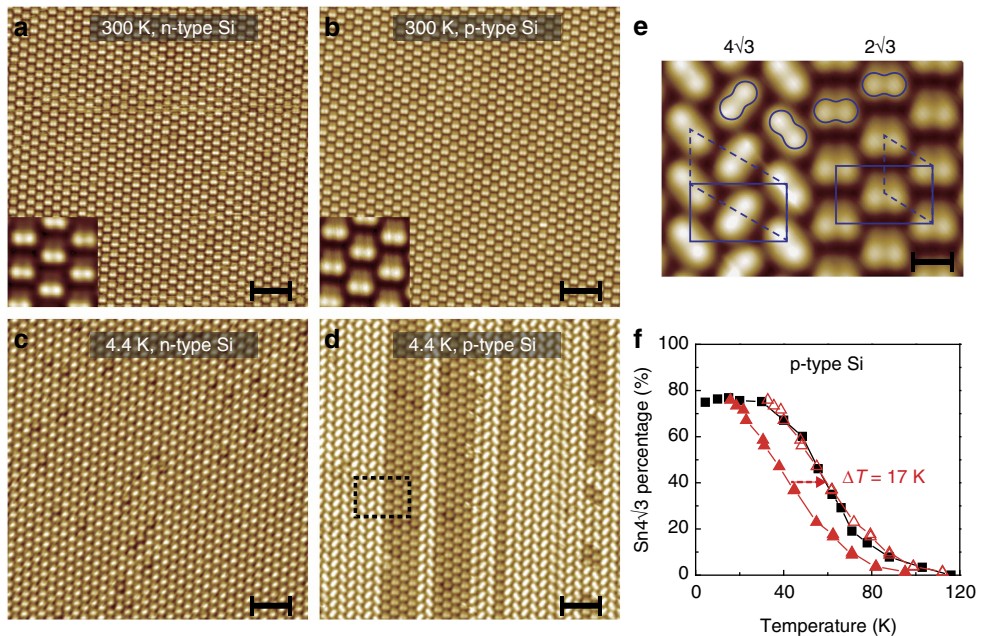

**Figure 2 | STM images of the $(2\sqrt{3} \times 2\sqrt{3})R30°$ Sn structures on n-type and p-type Si(111).** Notice the transition from the $(2\sqrt{3} \times 2\sqrt{3})R30°$ to $(4\sqrt{3} \times 2\sqrt{3})R30°$ phase for the p-type (B-$2\sqrt{3}$Sn) interface in **b**,**d**. No such transition is observed for the n-type interface (**a**,**c**). Scale bars in (**a**–**d**) are 5 nm; scale bar in **e** is 1 nm. Scanning parameters ($V_s$ and $I_t$) are 2 V, 0.2 nA (**a**), 1 V, 0.2 nA (**a** inset), 2 V, 0.5 nA (**b**), 2 V 0.3 nA (**b** inset), −2.3 V, 0.5 nA (**c**), +1.5 V, 0.02 nA (**d**) and +1.2 V, 0.15 nA (**e**). (**e**) Zoomed-in image of the surface area marked in **d**. Dashed lines mark the primitive $(2\sqrt{3} \times 2\sqrt{3})R30°$ and $(4\sqrt{3} \times 2\sqrt{3})R30°$ unit cells of the B-$2\sqrt{3}$Sn and B-$4\sqrt{3}$Sn phase, respectively, while solid lines mark the corresponding centred-rectangular and rectangular unit cells. Up-dimers are marked by ovals. (**f**) Area fractions of the B-$4\sqrt{3}$Sn (solid squares) and p-$4\sqrt{3}$Sn (solid triangles) surface phases as a function of temperature. To determine the change in transition temperature, the p-$4\sqrt{3}$Sn data were rigidly shifted and superposed with the B-$4\sqrt{3}$Sn data (hollow triangles). Each data point is obtained from a surface area that is at least 14,000 nm² in size, and is fully converged as a function of image size. Areas close to step edges and domain boundaries were excluded from the statistics.

two bond distances. There are four boron atoms per $(2\sqrt{3} \times 2\sqrt{3})R30°$ unit cell. The Sn layer is composed of 14 Sn atoms per unit cell with 4 Sn atoms adopting an outward relaxed double-dimer configuration. The presence of these tetramers gives the structure a double-layer appearance (see Supplementary Note 2). Room-temperature STM images in Fig. 2a,b reveal identical structures for the n-$2\sqrt{3}$Sn and B-$2\sqrt{3}$Sn interfaces. High-resolution insets reveal the characteristic double-dimer structure for both n-$2\sqrt{3}$Sn and B-$2\sqrt{3}$Sn. The two dimers are symmetric but differ in height, consistent with previous results[19–22]. The close similarity between the n-$2\sqrt{3}$Sn and B-$2\sqrt{3}$Sn structures is corroborated by the nearly identical $I(V)$ spectra in low-energy electron diffraction as well as their identical melting temperature[19] (see Supplementary Notes 3 and 4). Evidently, the subsurface boron-dopant layer does not affect the structure of the Sn overlayer in a significant way.

**Spontaneous symmetry breaking via hole doping.** Upon cooling to 4.4 K, the electron-doped n-$2\sqrt{3}$Sn surface remains fully intact (Fig. 2c). In contrast, the hole-doped B-$2\sqrt{3}$Sn surface gradually transforms into a new $(4\sqrt{3} \times 2\sqrt{3})R30°$ phase (henceforth B-$4\sqrt{3}$Sn) below ~100 K (Fig. 2d). Here, the bright dimers are rotated 45°, forming a staggered zigzag pattern (Fig. 2e). The formation of this structure appears to be 'tip assisted'. In particular, the freshly prepared surface is fully covered by the B-$2\sqrt{3}$Sn phase, even at 4.4 K, provided that images are recorded with negative sample bias. By switching the tunnelling conditions to positive bias, the majority of the surface transforms into the new B-$4\sqrt{3}$Sn structure. In contrast to observations on Si(100)[23], the tip-assisted formation of this hidden phase is mostly

irreversible with respect to the tunnelling conditions at 4.4 K (see Supplementary Note 5).

The area fractions of the coexisting $2\sqrt{3}$Sn and $4\sqrt{3}$Sn structures were determined as a function of temperature by scanning the same image repeatedly in dual bias mode until the domain ratios are fully stabilized (Fig. 2f, see Supplementary Note 5). These ratios are fully reproducible and reversible with temperature (within the errors of margin indicated below), and are independent of the phase-change history of the surface. This suggests that they represent an equilibrium configuration of the system. Interestingly, the area fraction or 'order parameter' of the B-$4\sqrt{3}$Sn phase no longer changes below 40 K and saturates at $0.75 \pm 0.07$. Very similar behaviour is observed for p-$2\sqrt{3}$Sn (Supplementary Note 6), except that the order-parameter curve has shifted to lower temperature by ~17 K (Fig. 2f), consistent with the nominally lower hole content for this interface (Supplementary Notes 1 and 6).

It should be noted that the concentration of the near-surface B dopants differs significantly for the p-$2\sqrt{3}$Sn and B-$2\sqrt{3}$Sn surfaces, as the latter contains 1/3 ML of boron at the $S_5$ lattice location. Yet, the B-$4\sqrt{3}$Sn area fractions at 4 K are nearly identical. It is therefore unlikely that the formation of the $4\sqrt{3}$Sn structure should be attributed to direct chemical interaction between boron and tin. Furthermore, there is no indication of striped inhomogeneity, either structurally or electronically, in the p-type $2\sqrt{3}$Sn systems that could trigger the formation of the striped $4\sqrt{3}$Sn domains. Such subsurface inhomogeneity would readily be evident in the $dI/dV$ maps of the $2\sqrt{3}$ structure (Supplementary Note 7). This then leaves hole doping as the only plausible mechanism for the formation of the $4\sqrt{3}$ domains. Interestingly, the exponential domain width distribution indicates

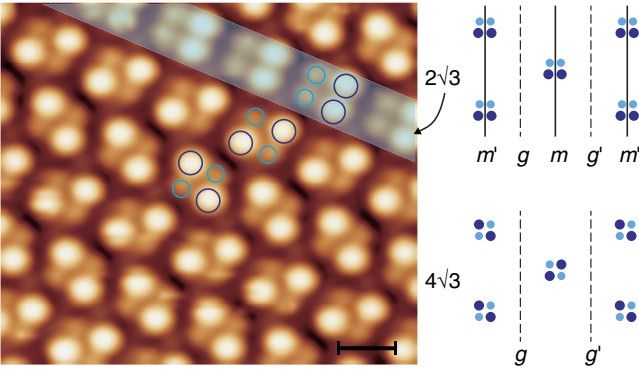

**Figure 3 | High-resolution STM image of the B-4√3Sn phase at 4.4 K.**
The green shaded area marks a narrow patch of the B-2√3Sn phase. The
Sn tetramers of the 2√3Sn and 4√3Sn structures are highlighted by circles.
Large circles mark the outward-relaxed atomic protrusions and smaller
circles the inward-relaxed atomic features. Scanning parameters are
$V_s = 1\,V$, $I_t = 0.2\,nA$. The scale bar is 1 nm. Mirror plane and glide plane
symmetries are indicated on the right.

that neither elastic nor electrostatic long-range interactions
play a significant role in stabilizing the unusual coexistence of
the 2√3 and 4√3 structures (Supplementary Note 8).

Figure 3 presents a high-resolution STM image of the B-4√3Sn
phase, along with a remnant patch of the B-2√3Sn phase,
recorded at 4.4 K. The tetramer subunits of the B-4√3Sn phase
acquire (approximate) twofold rotational symmetry, whereas the
tetramers of the B-2√3Sn phase exhibit mirror plane symmetry.
As indicated in the right panel of Fig. 3, the high-temperature
2√3 phase has a rhombic space group symmetry ($cm$, 2D space
group no. 5), characterized by the existence of both mirror ($m, m'$)
and glide plane ($g, g'$) symmetry. The 4√3 image in Fig. 3 has
a rectangular space group symmetry ($p2gg$, space group no. 8),
but because the underlying substrate lacks twofold rotational
symmetry, the overall symmetry of the 4√3 structure must be
$pg$ (space group no. 4; the weak breaking of $p2gg$ symmetry can be
observed under different tunnelling conditions, as shown in
Supplementary Note 9). The structural transition is characterized
by a loss of mirror plane symmetry, as well as a loss of centring
translation if one adopts a centred rectangular unit cell for
the 2√3 phase (Figs 3 and 2e). Most importantly, the
high-temperature phase cannot be obtained via superposition of
low-temperature domains or by thermal averaging of the
up–down positions of the adatoms of the low-temperature phase.
Thus, the 2√3 to 4√3 transition appears to be displacive in
nature, which is highly unusual as most other continuous surface
structural transitions are of the order–disorder type as a result of
dynamical fluctuations of the adatom heights[24,25] or diffusive
motion of the Sn adatoms[26]. Note that the 2√3 phase has polar
symmetry, meaning that it could support a macroscopic electrical
polarization parallel to the mirror plane. The 4√3 phase is
approximately nonpolar. This could affect the relative stability of
the 2√3 phase in the presence of an external electric field and,
consequently, the activation barrier for the tip-assisted phase
transformation. A more detailed discussion is provided in the
Supplementary Note 5.

**Band alignment and chemical potential.** The electronic
structure and structural transition are further characterized by
scanning tunnelling spectroscopy at 77 and 4.4 K (Fig. 4). The
normalized differential tunnelling conductance ($dI/dV/(I/V)$) is
approximately proportional to the local density of states at the

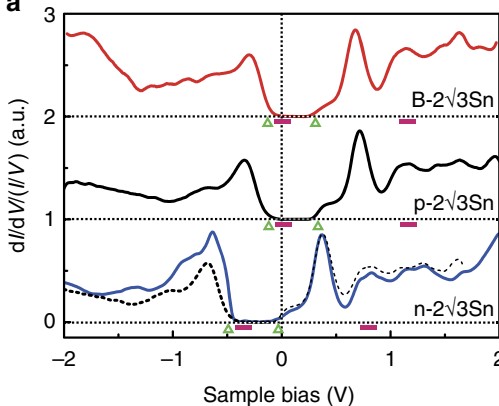

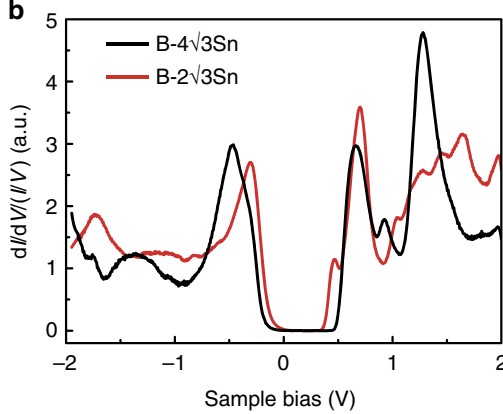

**Figure 4 | Scanning tunnelling spectroscopy (STS) measurements.**
(**a**) $dI/dV/(I/V)$ spectra of the B-2√3Sn, p-2√3Sn and n-2√3Sn surfaces
obtained at 77 K. The spectra for p-2√3Sn and B-2√3Sn are shifted
vertically for clarity. The Fermi level is located at 0 V. By shifting the
p-2√3Sn spectrum 0.36 eV toward lower energy (dashed line), and
superposing it with n-2√3Sn spectrum, it is seen that the low-energy
spectral features line-up almost perfectly. Triangles below each curve
mark the highest occupied surface state (left) and lowest unoccupied surface
state (right), while a solid bar indicates the estimated location of the
VBM (left) and CBM (right) of the bulk Si bands near the surface with
the bar length representing the error range (see Supplementary Note 1).
(**b**) $dI/dV/(I/V)$ spectra of the B-2√3Sn and B-4√3Sn interfaces obtained
at 4.4 K, showing a slightly larger band gap for the B-4√3Sn phase.
All STS spectra were measured on top of the bright dimers (STS at other
locations within a unit cell are showing similar features). Each $dI/dV/(I/V)$
spectrum is obtained by dividing the $dI/dV$ curve by its smoothed $I(V)$
curve[34].

surface. The $dI/dV/(I/V)$ curves for the n-2√3Sn, p-2√3Sn and
B-2√3Sn systems indicate an energy band gap of ∼0.45 eV. In
Supplementary Note 1, we find that the Fermi level is located
0.03 eV above the bottom of the empty surface state band for the
n-2√3Sn surface and ∼0.12 eV above the top of the filled surface
state band for the p-2√3Sn surface (77 K). Accordingly, the
maximum shift in the surface chemical potential achievable via
bulk doping amounts to 0.36 eV. Indeed, when shifting the
p-2√3Sn spectrum 0.36 eV to the left, as indicated by the dashed
line in Fig. 4a, we see that the spectral features of the n- and
p-type surfaces line up quite nicely. The B-2√3Sn and p-2√3Sn
spectra are very similar. The structural similarity between the
p- and n-type 2√3Sn surfaces and the negligible role of boron
chemistry in the formation of the 4√3Sn structure for the p-type
substrates strongly indicates that the stability of the mixed phase

is related to the location of the chemical potential at the surface. To further corroborate this point, we compensated the hole concentration by depositing 0.005 ML of K or Cs on the B-2$\sqrt{3}$Sn surface (see Supplementary Note 10 and Supplementary Fig. 12). This results in an upward shift of the chemical potential, consistent with the injection of n-type charge carriers from the alkali atoms, and an almost complete reversal of the 2$\sqrt{3}$Sn to 4$\sqrt{3}$Sn phase transformation at 60 K. This then conclusively shows that this phase transformation originates from modulation hole doping. The modulation doping approach works because Fermi-level pinning is weak, presumably due to the low density of defect states[27]. This allows for a rather wide sweep of the Fermi level location at the surface with relatively low carrier concentrations and provides access to different parts of the phase diagram.

The d$I$/d$V$ spectra of the p-type B-2$\sqrt{3}$Sn and B-4$\sqrt{3}$Sn phases (Fig. 4b) reveal some important differences as well, such as the peaks at $+1.2$ and $-0.5$ eV and the wider band gap of $\sim 0.7$ eV for the B-4$\sqrt{3}$Sn structure. The latter seems consistent with the fact that the B-4$\sqrt{3}$Sn structure is favoured at low temperature.

## Discussion

The formation of the mixed phase clearly requires nonthermal activation through a negatively biased STM tip. As such, it represents a hidden state of matter, although one that is long-lived and stabilized in the presence of hole doping. Whether or not the mixed phase represents a global minimum cannot be decided from the current investigation, though it is natural to assume that the broken symmetry state represents the total energy minimum. Nonetheless, exceptions to this rule exist[28].

The presence of a displacive insulator-to-insulator transition on a semiconductor surface is also highly unusual. While a continuous $cm \rightarrow pg$ phase transition would be symmetry allowed[29], the persistent coexistence of the 2$\sqrt{3}$Sn and 4$\sqrt{3}$Sn phases and crossover behaviour at high temperature are both indicative of a preempted first-order mechanism[30], one that requires activation by the STM tip. The latter may be needed to access a transitional and possibly symmetric tetramer configuration via local excitation. The subsequent vertical relaxations of the tetramer atoms and the doubling of the lattice periodicity would involve one or several soft lattice modes. Such a phonon instability and associated charge fluctuation is likely affected by the presence of screening charges at the interface with presumably opposite effects for the depletion layer and hole reservoir present on the n- and p-type Si substrates, respectively.

Finally, we note that surface defects do not play a significant role in the nucleation and pinning of the surface domains. All the evidence seems to indicate that the hidden phase is intrinsically phase separated, suggesting the presence of multiple minima in the thermodynamic potential as a function of carrier doping. This is reminiscent of the electronic phase separation in some complex oxides[31]. An important distinction here is that the two surface phases are both nonmetallic and that the domains are easily visualized.

In summary, a modulation doping approach, which introduces charge only but avoids chemical perturbation to the surface, allows access to novel surface phases and hidden symmetries. It is achieved by adjusting the bulk chemical potential and the concentration of subsurface B dopants. This method is expected to be applicable to other well-ordered semiconductor surface reconstructions as long as the corresponding surface states are located inside the bulk band gap, and as long as the substrate material can be doped both n- and p-type over a considerably wide range of doping levels. The same idea can be applied to complex oxide heterostructures.

It should therefore be possible to map out electronic phase diagrams in a variety of other surface systems. A most intriguing possibility would be the existence of a superconducting instability or quantum critical point in a surface layer[32]. Besides this conceptual appeal, we have also shown that cooperative phenomena can be realized and tuned in relatively simple $sp$-bonded 2D materials systems on a Si platform. This notion may ultimately bring the rich physics of complexity closer to the realm of technological innovation.

## Methods

**Preparation of the substrates.** Four types of Si(111) wafers were used: heavily As-doped n-type Si with room-temperature resistivity of 0.002 $\Omega$ cm, and three different B-doped p-type Si wafers with room temperature resistivities of $\sim 0.004$, $\sim 0.009$ and $\sim 0.03 \Omega$ cm. The heavily doped Si substrates give rise to finite conductivity at low temperatures, which is essential for the STM/scanning tunnelling spectroscopy measurements. Samples cut from these wafers were ultrasonically cleaned with acetone and alcohol, and then processed following standard procedures in ultrahigh vacuum: degassing at 600 °C overnight, flashing at $\sim 1,250$ °C and cooling from 900 °C to room temperature over the course of several minutes. A Si(111)-(7 × 7) reconstruction surface is obtained for the n-type and $\sim 0.03 \Omega$ cm p-type Si wafers. A ($\sqrt{3} \times \sqrt{3}$)$R30°$-B reconstructed surface is prepared on the most heavily B-doped (0.004 $\Omega$ cm) material by using a longer annealing time, for example, 5 h at 1,150 °C, followed by slow cooling to room temperature over the course of several hours to facilitate the segregation of $\sim 1/3$ ML of B to the $S_5$ sites below the surface[18]. A reduced annealing time for this heavily doped material produces a highly defective ($\sqrt{3} \times \sqrt{3}$)$R30°$-B surface with reduced boron content. The slightly more resistive 0.009 $\Omega$ cm wafer exhibits a mixed defective ($\sqrt{3} \times \sqrt{3}$)$R30°$-B and 7 × 7 surface[33].

**Sample preparation and measurements.** The substrates were thoroughly characterized with STM to ensure that well-ordered reconstructed surfaces were present before the Sn deposition. The ($\sqrt{3} \times \sqrt{3}$)$R30°$-B surface contains a small number of Si$_B$ defects where Si atoms populate some of the $S_5$ lattice sites[18]. For studies of the B-2$\sqrt{3}$Sn phase, we kept the concentration of Si$_B$ defects below $\sim 15$ defects per 2,500 nm$^2$. The Si(111)(2$\sqrt{3}$ × 2$\sqrt{3}$)$R30°$-Sn surface reconstructions were prepared on the four different substrates by thermally depositing $\sim 1$ ML of Sn at a substrate temperature of $\sim 550$ °C. The sample was post annealed at this temperature for several minutes and cooled to room temperature.

For room-temperature and low-temperature measurements, we used a commercial low-temperature STM (Omicron). Chemically etched tungsten tips were used and checked on gold films before the measurements. This system is also equipped with potassium and caesium dispensers from SAES. The potassium or caesium atoms are deposited onto the surfaces held at $\sim 100$ K to reduce the diffusion of the adsorbed atoms. The d$I$/d$V$ signal was acquired by a lock-in amplifier using a typical frequency of 831 Hz and modulation amplitude between 1 and 10 mV. Low-energy electron diffraction (for structural characterization) and X-ray photoelectron spectroscopy (XPS) studies (for band bending measurements) were conducted in a separate UHV system. This system was also equipped with a Sn effusion cell and a variable-temperature STM so that the surface preparation and atomic-scale characterization could be reproduced. The low-energy electron diffraction and X-ray photoelectron spectroscopy measurements are described in Supplementary Notes 1, 3 and 4.

**Data availability.** Original data will be made available upon request (hanno@utk.edu).

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

## Acknowledgements

We thank Seho Yi and Jun-Hyung Cho for many stimulating discussions. This work was funded by the National Science Foundation under Grant No. DMR 1410265. G.L. acknowledges supports from the National Research Foundation of Korea (NRF) funded by the Korean government (MSIP) (No. 2015001948).

## Author contributions

F.M., D.M., W.T. and T.S.S. did the experiments; P.V. provided technical support for the experiments; Y.-T.H. and R.D.D. analysed the low-energy electron diffraction $I(V)$ data; P.C.S. and H.H.W. led the project; P.C.S., H.H.W., F.M., D.M. and T.S.S. analysed other data and wrote the manuscript; and all authors were involved in discussion.

## Additional information

**Competing financial interests:** The authors declare no competing financial interests.

