## [Peer Review File · Nature Communications]

Reviewers' comments:

Reviewer #1 (Remarks to the Author):

The authors present results on the $\text{Si}(111)(2\sqrt{3}\times 2\sqrt{3})R30^\circ$ -Sn surface reconstruction using both n-type and p-type doped substrates. The authors claim that the $(2\sqrt{3}\times 2\sqrt{3})R30^\circ$ surface reconstruction retains its translational symmetry upon electron doping, but the hole-doped reconstruction phase-separates into insulating $(2\sqrt{3}\times 2\sqrt{3})R30^\circ$ and $(4\sqrt{3}\times 2\sqrt{3})R30^\circ$ domains. The formation mechanism of this novel $(4\sqrt{3}\times 2\sqrt{3})R30^\circ$ phase is explained with a rare displacive transition, assisted by proximity coupling to the tip of a scanning tunneling microscope (STM).

I reject the paper for various reasons: 1) I consider that this subject is not of a general interest. The argument that depends on the doping seems too loose (it looks to appear to any doping!! but maybe at different temperatures, 2) If doping is not now the main argument the system has no interest. 3) It is therefore a new phase transition, of not much interest from my point of view. 4) The influence of STM, recognized and discussed by the authors, although it was a new and interesting system makes the importance of observations very limited: this is another surface phenomenon (as many of those in the literature) influenced by the STM!

I consider that the paper is not suitable for Nature Communications.

Note: On figure 1 is presented for the $(2\sqrt{3}\times 2\sqrt{3})R30^\circ$ phase, the structural model proposed by "Tornevik" while this structure is nowadays unclear and still under dispute.

Reviewer #2 (Remarks to the Author):

The paper "Hidden phase in two-dimensional Sn layer stabilised by modulation hole doping" reports an extensive study of the structural and electronic properties of an ordered phase of Sn on $\text{Si}(111)$ surface.

The $2\sqrt{3}\times 2\sqrt{3}$ phase is a well characterised surface and well characterised in the past. STM (ref 16 and 19 of the manuscript), XPS and STS (Surf. Sci. 554, 109 (2004) to be added in the reference list), and ARPES (ref.19) studies were performed in the past, but low temperature phase transitions were not reported.

The original approach of the authors resides in the use of different substrates with different degree (and sign) of dopants.

Low temperature STM images revealed a (low temperature) phase transition in the case of Sn deposited on p-type doped substrate.

Boron doped Silicon substrate assure high degree of doping and increase the transition temperature.

The phase transition is reported to be tip induced and reproducible.

Although, in the present form, a complete understanding of the physical mechanisms driving the phase transition are almost lacking, I judge the experimental report robust, interesting and deserving publication.

This experimental demonstration of selective doping can open interesting research lines in surface physics.

Reviewer #3 (Remarks to the Author):

In the manuscript "Hidden phase in a two-dimensional Sn layer stabilized by modulation hole doping", the authors propose that they have performed modulation doping on a Sn reconstruction on top of a silicon substrate. Modulation doping means that doping is performed without significantly affecting the structure of the layer giving rise to the main physical properties of the system. If the results were true and the procedure could be generalized to other systems in surface science, interesting perspectives could appear in correlated surfaces. Moreover, a new symmetry is observed on the surface upon scanning with the STM tip, the "hidden phase". When doping the silicon substrate with boron (hole-doping), the initial $(2\sqrt{3}\times 2\sqrt{3})R30^\circ$ surface may exhibit stripe domains with a $(4\sqrt{3}\times 4\sqrt{3})R30^\circ$ symmetry due to the field of a negatively-biased STM tip. The authors believe that this phase is related to the hole doping and is a displacive transition, instead of the more usual vertical fluctuation of the atomic position of the adatoms.

The manuscript should be more clear in the arguments favoring the modulation doping. In particular, it should be justified that the structure of the doped regions is not modified. For instance, what is the proof that the $4\sqrt{3}$ is only driven by the doping and not by an accumulation of B atoms along stripes? Does the domain size of the $4\sqrt{3}$ domains vary with doping? Also some discussion about the possible mechanisms giving rise to the modulation doping in this system should be presented. Would it be possible to generalize the mechanism to other surfaces?

The $4\sqrt{3}$ reconstruction is explained in terms of a displacive transition that is presented as 'rare'. It is indeed the case, and has been previously observed on Sn tetramers on a similar $2\sqrt{3}$ reconstruction (ref. 22). The displacive character is thus probably a more general mechanism of Sn reconstructions with different coverage on hole-doped silicon. Such an issue is not very well described.

Finally, I have a minor question, concerning the spatial location where the STS spectra of the fig. 4 were recorded. It will be helpful to know if they are averaged on the whole unit cell or they correspond at a particular location, which could render the comparison between the different systems less relevant.

In conclusion, the manuscript presents promising statements which can open new perspectives on the field of correlated surfaces. However, these new statements should be better justified before publication, as I stated in my above comments.

Detailed responses to every point raised by the referees.

(NCOMMS-16-20371 by Ming, *et al.*)

Comments by Referee 2

“The paper "Hidden phase in two-dimensional Sn layer stabilised by modulation hole doping" reports an extensive study of the structural and electronic properties of an ordered phase of Sn on Si(111) surface.

The $2\sqrt{3}\times 2\sqrt{3}$ phase is a well characterised surface and well characterised in the past. STM (ref 16 and 19 of the manuscript) , XPS and STS (Surf. Sci. 554, 109 (2004) to be added in the reference list), and ARPES (ref.19) studies were performed in the past , but low temperature phase transitions were not reported.

The original approach of the authors resides in the use of different substrates with different degree (and sign) of dopants.

Low temperature STM images revealed a (low temperature) phase transition in the case of Sn deposited on p-type doped substrate.

Boron doped Silicon substrate assure high degree of doping and increase the transition temperature.

The phase transition is reported to be tip induced and reproducible.

Although, in the present form, a complete understanding of the physical mechanisms driving the phase transition are almost lacking, I judge the experimental report robust, interesting and deserving publication.

This experimental demonstration of selective doping can open interesting research lines in surface physics.”

Response: We thank the referee for his/her very positive assessment of our work. Referee 2 advised publication without changes (other than inserting one particular reference). The referee correctly pointed out that complete understanding of the physical mechanisms of the $2\sqrt{3}$ to $4\sqrt{3}$ transition is lacking. Nonetheless, he or she deemed the experimental results robust and interesting enough to merit publication because *“This experimental demonstration of selective doping can open interesting research lines in surface physics”*. Note that a full understanding of the physical mechanism for the insulator-to-insulator transition requires unambiguous determination of the atomic structure and in-depth theoretical calculations and analysis. Given the very large size of the unit cell, this would be a major undertaking beyond the scope of the current manuscript. The key issue here is the emergence of a hidden phase of matter on a simple

semiconductor surface in response to modulation hole-doping. We fully expect that these results will inspire further studies of the structure and driving mechanism of the transition.

We have added *Surf. Sci.* 554, 109 (2004) to our reference list.

Comments by Referee 3

“In the manuscript “Hidden phase in a two-dimensional Sn layer stabilized by modulation hole doping”, the authors propose that they have performed modulation doping on a Sn reconstruction on top of a silicon substrate. Modulation doping means that doping is performed without significantly affecting the structure of the layer giving rise to the main physical properties of the system. If the results were true and the procedure could be generalized to other systems in surface science, interesting perspectives could appear in correlated surfaces. Moreover, a new symmetry is observed on the surface upon scanning with the STM tip, the “hidden phase”. When doping the silicon substrate with boron (hole-doping), the initial $(2\sqrt{3}\times 2\sqrt{3})R30^\circ$ surface may exhibit stripe domains with a $(4\sqrt{3}\times 4\sqrt{3})R30^\circ$ symmetry due to the field of a negatively-biased STM tip. The authors believe that this phase is related to the hole doping and is a displacive transition, instead of the more usual vertical fluctuation of the atomic position of the adatoms.”

Response: We thank the reviewer for recognizing the importance and broad appeal of this work.

“The manuscript should be more clear in the arguments favoring the modulation doping. In particular, it should be justified that the structure of the doped regions is not modified. For instance, what is the proof that the $4\sqrt{3}$ is only driven by the doping and not by an accumulation of B atoms along stripes? Does the domain size of the $4\sqrt{3}$ domains vary with doping”

Response: We agree with the referee that the manuscript did not contain a proper exclusion of subsurface dopant ordering as a potential cause for the $4\sqrt{3}$ Sn stripe formation. We are now able to exclude this hypothesis, and addressed this issue on page 7 of the revised manuscript. In addition, we added a new Supplementary Note (#7) containing new data that exclude subsurface dopant ordering as the origin of $4\sqrt{3}$ stripe formation. Our arguments are based on the following observations:

- (a) In the original submission, we excluded a chemically induced structure modification of the $p\text{-}2\sqrt{3}\text{Sn}$ and $B\text{-}2\sqrt{3}\text{Sn}$ structures due to boron incorporation. This exclusion was based on the nearly identical LEED I(V) spectra for the $2\sqrt{3}$ phases on the n- and p-type substrate, their nearly identical melting temperature, and their exactly-identical appearance in STM images. The only real difference appears to be the location of the chemical potential. Of course, it is possible that the structure modifications are too small to be detected by the above techniques. However, such a structure modification would by far be the largest for the $(\sqrt{3}\times\sqrt{3})\text{-B}$ substrate with the $1/3$ ML boron

underlayer. (The subsurface boron concentration for the 7×7 substrates is far less). It would then be very difficult to rationalize why the $4\sqrt{3}$ domains are striped with a width distribution that is nearly identical for the $(\sqrt{3}\times\sqrt{3})$ -B and 7×7 substrates (see below). Instead, we argue that the hole doped surface phase is inherently phase separated into striped domains where the stability of the striped phase is determined by the doping level. This explanation clearly is much more plausible.

- (b) If the boron atoms were to accumulate into striped patches, we should expect to see lateral variations in the electronic properties. The patches would be very heavily hole-doped while regions in between the patches would be relatively depleted of holes. This, in turn, should give rise to lateral variations in the tunneling conductance. To check this possibility, we collected dI/dV maps (or conductance maps) of the $2\sqrt{3}$ surface, measured at 77K, *i.e.*, just above the transition temperature to the $4\sqrt{3}$ structure. The tunneling bias is -0.4 eV for p-type and -0.75 eV for n-type samples. At this bias, the tunneling states are degenerate with the bulk valence band continuum and, consequently, we should be more sensitive to the subsurface region. Importantly, our dI/dV maps show no evidence of stripe-like patterns in the tunneling conductance, and hence a stripe-like aggregation of subsurface boron can be excluded. These new data are presented in Supplementary Note 7.

There is some disorder/inhomogeneity in the dI/dV maps of the $2\sqrt{3}$ surface, grown on the $(\sqrt{3}\times\sqrt{3})$ -B substrate. However, this inhomogeneity is not seen for the lightly doped p-type or n-type substrates. Because the striped $4\sqrt{3}$ structure appears on all hole doped substrates, this subsurface inhomogeneity is not correlated with the appearance of the striped $4\sqrt{3}$ Sn structure. The inhomogeneity in the dI/dV maps of the B- $2\sqrt{3}$ Sn structure can be attributed to inhomogeneity in the distribution of boron atoms below the surface (J. Phys. Chem. C 2014, 118, 15744–15753). This is confirmed by the absence of disorder in constant current STM images of the surface atomic structure that were recorded simultaneously and at the same tunnel bias (Supplementary Fig. 8).

- (c) We also compared the domain size distribution of low-temperature $4\sqrt{3}$ Sn structure for different hole concentrations (updated Supplementary Fig. 9 with new data for the p- $2\sqrt{3}$ Sn and p- $4\sqrt{3}$ Sn structures). The surfaces grown on p-type Si(111) 7×7 (with low boron concentration beneath the surface) and on the Si(111) $(\sqrt{3}\times\sqrt{3})$ -B surface (with the maximum amount of B near the surface, including $1/3$ ML of boron at the S5 lattice locations) show a very similar distribution of the $4\sqrt{3}$ Sn domain widths. Also the $4\sqrt{3}$ coverage measured at 5 K, is identical. This makes it extremely unlikely that the $4\sqrt{3}$ Sn domain formation should be attributed to a hypothetical self-organization of different boron concentrations into very similar stripe-like patterns. The only plausible explanation is that the striped phase is formed due to the hole doping of our system, where the stability of the striped phase depends on the doping level.

We have made following revisions in the new paper to present these points:

On Page 7, we have added the following discussion to exclude structural modification as the origin for the formation of the mixed phase.

“It should be noted that the concentration of the near-surface B dopants differs significantly for the $p\text{-}2\sqrt{3}\text{Sn}$ and $\text{B-}2\sqrt{3}\text{Sn}$ surfaces, as the latter contains $1/3$ ML of boron at the S_5 lattice location. Yet, the $\text{B-}4\sqrt{3}\text{Sn}$ area fractions at 4 K are nearly identical. It is therefore unlikely that the formation of the $4\sqrt{3}\text{Sn}$ structure should be attributed to direct chemical interaction between boron and tin. Furthermore, there is no indication of striped inhomogeneity, either structurally or electronically, in the p-type $2\sqrt{3}\text{Sn}$ systems that could trigger the formation of the striped $4\sqrt{3}\text{Sn}$ domains. Such subsurface inhomogeneity would readily be evident in the dI/dV maps of the $2\sqrt{3}$ structure (Supplementary Note 7). This then leaves hole doping as the only plausible mechanism for the formation of the $4\sqrt{3}$ domains”.

We added a new Supplementary Note 7, where we discuss the new dI/dV maps for different doping levels (presented in Supplementary Fig. 8). We furthermore acquired statistical data for the domain width distribution and presented those in the updated Supplementary Figure 9.

“Also some discussion about the possible mechanisms giving rise to the modulation doping in this system should be presented.”

Response: We thank the reviewer for his or her most pertinent comment. We admit that we could have been more explicit about the modulation doping mechanism. The argument primarily revolves around the physics of semiconductor space charge layers. While some of this was discussed in Supplementary Note 8 of the original manuscript, we now address this early on in the paper and provide a more in-depth discussion in the new Supplementary Note 1. The effort also includes a new Supplementary Fig. 2 to show the $I(V)$ curve fittings for *quantitative* determining the surface band gap and surface state band edge locations, and an updated Fig. 4a to clearly presents the band alignment of the surface state and bulk band edges.

On page 5, we inserted the following paragraph:

“As discussed in Supplementary Note 1, any change in the bulk chemical will lead to a readjustment of the chemical potential in the surface layer. This readjustment involves charge transfer between the surface states and the bulk, and generally implies doping by the excess charges or ‘dopant charges’ in the surface states. The resulting band alignments for the n- and p-type systems are shown in Supplementary Fig. 1.”

Would it be possible to generalize the mechanism to other surfaces?”

This mechanism is directly applicable to other surface reconstructions on Si, and other semiconductor surfaces where the bulk Fermi level could be adjusted by dopants. The formation of a space charge layer below a semiconductor surface is very common (see Mönch, W. *Semiconductor surfaces and interfaces*). There will always be a charge transfer between the

bulk substrate and the donor or acceptor levels at the surface, due to the intrinsic mismatch in Fermi level position prior to establishing electrical and thermodynamic ‘contact’ between the surface and the bulk. If the donor or acceptor levels at the surface are simply defect levels, then there is not much interest. However, many atomically clean and well-ordered surfaces (with preferably few defect levels) exhibit a well-defined surface state spectrum where the surface states are located inside the bulk band gap and, consequently, the Fermi level at the surface is determined to zeroth order by the electron count in the surface states. (The latter is determined by the structure and composition of the surface). However, as we discussed, the Fermi level at the surface must line up with the Fermi level in the bulk, and the equilibration of the Fermi levels involves a charge transfer between the surface and the bulk, meaning that we are adding electrons or holes to the surface state bands. This is a well-known phenomenon. What makes our work unique is that we recognized that we could exploit this phenomenon to explore the stability of surface phases as a function of carrier density in the surface states, similar to the case of modulation doping in semiconductor heterostructures and cuprate superconductors, or gating of 2D materials. This is the first systematic study exploring the modulation doping concept for 2D surface state systems. The observation of a new phase whose stability depends on the doping level shows that our approach works. There is every reason to believe that the method will work for other semiconductors surfaces.

Even more generally speaking, buried delta-doped layers have recently been applied to access high doping levels in complex oxides (Science 326, 699–702 (2009)). As such, our approach is certain to impact a broad array of surface systems or other 2D material systems that are deposited or exfoliated onto a substrate.

In the conclusion part on pages 10 and 11, we inserted the following sentence to indicate the broad applicability of the modulation doping method:

“This method is expected to be applicable to other well-ordered semiconductor surface reconstructions as long as the corresponding surface states are located inside the bulk band gap, and as long as the substrate material can be doped both n- and p-type over a considerably wide range of doping levels. The same idea can be applied to complex oxide heterostructures.”

“The $4\sqrt{3}$ reconstruction is explained in terms of a displacive transition that is presented as ‘rare’. It is indeed the case, and has been previously observed on Sn tetramers on a similar $2\sqrt{3}$ reconstruction (ref. 22). The displacive character is thus probably a more general mechanism of Sn reconstructions with different coverage on hole-doped silicon. Such an issue is not very well described.”

Response: The paper “Phys. Rev. Lett. 114, 196101 (2015)” (Original Ref. 22), describes a temperature driven phase transition from a $(2\sqrt{3}\times 2\sqrt{3})R30^\circ$ structure to a $(\sqrt{3}\times \sqrt{3})R30^\circ$ at $\sim 520\text{K}$. It was argued that the Sn atoms exhibit a diffusive motion among 24 equivalent configurations above the transition temperature, and was recognized to be an order-disorder transition in this reference (first page, middle of right column). There is a fundamental distinction between an order-disorder transition, due to a thermal averaging mechanism, and a

purely displacive transition between two ordered states. The latter usually involves a soft phonon anomaly. Clearly, thermal averaging of the adatom positions of the $4\sqrt{3}\text{Sn}$ phase would never produce the $2\sqrt{3}\text{Sn}$ phase. We addressed this issue as follows (page 8):

“Thus, the $2\sqrt{3}$ to $4\sqrt{3}$ transition appears to be displacive in nature, which is highly unusual as most other continuous surface structural transitions are of the order-disorder type as a result of dynamical fluctuations of the adatom heights (24, 25) or diffusive motion of the surface atoms (26).”

“Finally, I have a minor question, concerning the spatial location where the STS spectra of the fig. 4 were recorded. It will be helpful to know if they are averaged on the whole unit cell or they correspond at a particular location, which could render the comparison between the different systems less relevant.”

Response: The STS curves shown in Fig. 4 are all taken on the bright dimers, both for $2\sqrt{3}\text{Sn}$ and $4\sqrt{3}\text{Sn}$ phase. We did compare them to area averaged STS, where all locations in the unit cell contribute to the STS data, and found that all the major features, including the gap and peak positions, are very similar. Therefore, the STS data presented in the paper provide a valid comparison of $4\sqrt{3}\text{Sn}$ and $2\sqrt{3}\text{Sn}$ phases in Fig. 4b. We added following description in the caption of Fig. 4 to indicate the STS location:

“All STS spectra were measured on top of the bright dimers (STS at other locations within a unit cell are showing similar features).”

“In conclusion, the manuscript presents promising statements which can open new perspectives on the field of correlated surfaces. However, these new statements should be better justified before publication, as I stated in my above comments.”

Response: We thank the referee’s overall positive assessment on our work. His or her comments helped us a great deal in reinforcing the experimental evidence for modulation doping and in better formulating the broader impact of our work. We believe all the issues raised by this referee are well settled.

Comment by Referee 1

“I reject the paper for various reasons: 1) I consider that this subject is not of a general interest. The argument that depends on the doping seems too loose (it looks to appear to any doping!! but maybe at different temperatures,”

Response: We thank the reviewer for his or her effort. The reviewer appears to state that the subject is not of general interest because the doping argument is “too loose”. Unfortunately, the reviewer does not offer clear arguments or specific details on how he/she reaches this conclusion. The reviewer does seem to notice a correlation between bulk doping level and the

temperature dependent stability of the $4\sqrt{3}$ phase, but appears to conclude this correlation is not relevant for the observed doping-dependent properties. We wish to point out that this correlation is precisely the main point of our paper: it clearly supports our conclusion that the observed phase transition is due to, and dependent on, modulation doping as imposed on the surface by the substrate chemical potential. Indeed, the structural transition temperature on p-type substrates depends on the doping level, and on the n-type substrates where the hole doping goes to zero, there is no transition at the experimentally accessible temperatures (i.e. the transition temperature is <3 K). Insofar the reviewer might have been thinking along the same lines as Referee 3, we refer to our response to Referee 3 where we clarify our conclusions.

“2) If doping is not now the main argument the system has no interest.3) It is therefore a new phase transition, of not much interest from my point of view.”

Response: Comments (2) and (3) were based on the reviewer’s supposition that doping is not the cause of the phase transition in comment (1). Since the evidence for doping is very strong, as mentioned in our response to comment (1) and as extensively discussed in our response to Referee 3, we are confident that the referee will agree that the paper is, in his or her own words, indeed of ‘much interest’.

“4) The influence of STM, recognized and discussed by the authors, although it was a new and interesting system makes the importance of observations very limited: this is another surface phenomenon (as many of those in the literature) influenced by the STM!”

Response: We disagree with the referee’s judgment that the reported tip-assisted phase formation mechanism is like “many of those in the literature”. (Again, the reviewer does not provide any specific detail or reference). We quote our statement in the Supplementary Note 5: “STM induced phase transformations have been seen in other systems (13-18). However, all of them turned out to be reversible, even below 10 K (13, 14, 17), whereas here, the conversion is irreversible and the B-4 $\sqrt{3}$ Sn configuration is static. Moreover, none of these systems involve a transition between polar and non-polar symmetries.” Instead of being just another tip effect, the crucial point is that we have accessed a long-lived hidden phase through a combination of modulation doping and STM tip assist. We not only accessed the hidden state but also determined its stability as a function of the hole density. This is what makes the work truly unique. The irreversibility of the phase transformation under the STM tip is absolutely essential here.

“Note: On figure 1 is presented for the $(2\sqrt{3}\times 2\sqrt{3})R30^\circ$ phase, the structural model proposed by “Tornevik” while this structure is nowadays unclear and still under dispute.”

Response: While we do recognize that recently a competing structural model has been proposed for the $2\sqrt{3}$ surface, cited in Ref. 26, we remark that our LEED I(V) data, STM evidence, and XPS coverage determination are inconsistent with this model. Moreover, we have compared several models in the literature with our LEED I(V) data analysis and find that the

Tornevik model agrees best with the data. While this may not be the final structure, it is the best candidate structure available at this point in time. We have added this remark in the caption of Fig. 1. The LEED I(V) work will be the subject of a future paper. It should be noted that the structure presented in Fig. 1, while based on the Tornevik model, is only for illustrative purposes, and the specific model does not influence the generality of our results.

Reviewers' Comments:

Reviewer #1 (Remarks to the Author)

In this work, the authors have combined several experimental techniques as STM, XPS and LEED to investigate the phase transition of $B-2\sqrt{3}$ to $4\sqrt{3}$ in a Silicon sample highly doped with Boron. This finding is reasonable and very important for doped-silicon research, will definitely promote further attempts to synthesize analogous surface phase transition with other dopants.

This new version has clarify some questions but comments need to be further considered:

1-The Supplementary Fig.6 is relevant on the publication where can be observed how the STM tip assist the phase transformation from the $B-2\sqrt{3}$ to the $4\sqrt{3}$ when switching from negative to positive sample bias. Similar figures should be included for the $n-2\sqrt{3}$ and $p-2\sqrt{3}$ samples at 4.4K to see the differences.

The comparison will evidence at Low Temperature the doping effect by switching bias.

2-Figures shown on the paper for n-type samples recorded at low temperature are taken with negative bias while authors indicates the relevance of switching to positive sample bias to observe or not the phase transition. The STM images with positive bias should be included.

3-If dopants are the responsible of the phase transition certainly on the presented data are lacking conclusive experiments where for instance the dopant is changed i.e. on the p-samples use one sample highly doped with Gallium for instance and see if you also obtain the phase transition!

In conclusion, the manuscript presents promising results that can open new perspectives if finally confirmed. However, still these conclusions should be better experimentally justified before publication.

Reviewer #2 (Remarks to the Author)

The authors have answered to all the questions and comments raised by the referees in a satisfactory way.

I do not find any strong obstacles to the publication of the paper in the present form.

Reviewer #3 (Remarks to the Author)

Dear Dr. Milana,

The authors have thoroughly and precisely answered to my comments. They have introduced the necessary modifications in the manuscript accordingly. In my opinion, the manuscript deserves now publication in Nature Communications.

Detailed responses to every point raised by the referees.

(NCOMMS-16-20371A-Z by Ming, *et al.*)

Comments by Referee 1

“In this work, the authors have combined several experimental techniques as STM, XPS and LEED to investigate the phase transition of $B-2\sqrt{3}$ to $4\sqrt{3}$ in a Silicon sample highly doped with Boron. This finding is reasonable and very important for doped-silicon research, will definitely promote further attempts to synthesize analogous surface phase transition with other dopants.”

Response: We thank the referee’s positive assessment on the importance of our paper.

“This new version has clarify some questions but comments need to be further considered:

1-The Supplementary Fig.6 is relevant on the publication where can be observed how the STM tip assist the phase transformation from the $B-2\sqrt{3}$ to the $4\sqrt{3}$ when switching from negative to positive sample bias. Similar figures should be included for the $n-2\sqrt{3}$ and $p-2\sqrt{3}$ samples at 4.4K to see the differences. The comparison will evidence at Low Temperature the doping effect by switching bias.

2-Figures shown on the paper for n-type samples recorded at low temperature are taken with negative bias while authors indicates the relevance of switching to positive sample bias to observe or not the phase transition. The STM images with positive bias should be included.”

Response: We have added STM images (added to Supplementary Fig. 6) for the $p-2\sqrt{3}\text{Sn}$ surface at 5 K to demonstrate that the tip-assisted phase transformation is present also for the moderately doped p-type sample. For the $n-2\sqrt{3}\text{Sn}$ surface, we present a positive bias STM image measured at 5 K (new Supplementary Fig. 7), demonstrating that the $4\sqrt{3}\text{Sn}$ phase does not appear on this sample regardless of the scanning bias. In Supplementary Note 5, we added following remarks:

“Very similar tip-assisted $4\sqrt{3}\text{Sn}$ formation behavior is observed for the $p-2\sqrt{3}\text{Sn}$ surface at 5 K: the fresh surface is fully covered by $p-2\sqrt{3}\text{Sn}$ phase if scanned at negative bias (panel (e)). Upon switching to positive bias (panel (f)), most the surface will transform to the $p-4\sqrt{3}\text{Sn}$ phase. In contrast, the electron doped Sn surface always show a pure $2\sqrt{3}\text{Sn}$ phase, regardless of the scanning parameters (see STM images at 5 K with negative bias in Fig. 2c and with positive bias in Supplementary Fig. 7).”

3-If dopants are the responsible of the phase transition certainly on the presented data are lacking conclusive experiments where for instance the dopant is changed i.e. on the p-samples use one sample highly doped with Gallium for instance and see if you also obtain the phase transition!

Response: The referee asks for additional experiments to further verify the connection between hole doping and the new phase transition, and explicitly proposed an experiment by using a p-type Si substrate that is not boron doped. Unfortunately, it is not possible for us to perform the proposed experiment because we could not find any manufacturer selling p-type single crystal polished Si(111) wafers (other than boron doped wafers) that have a doping concentration (and conductivity) *high enough to allow low temperature STM experiments*. A potential reason for the lack of heavily doped wafers with e.g. Ga or Al dopants on the market is mostly because these dopants have lower solubility and much smaller segregation coefficient in silicon than boron (see Bell System Technical Journal 39 205-233 (1960)). Nevertheless, we have conducted an alternative experiment to *conclusively* verify that hole-doping is indeed the root cause of the observed tip-induced phase transformation. We have doped electrons to the modulation hole-doped Sn surface reconstructions via alkali atom adsorption. The electron doping from alkali atoms adsorbed on the p-type Sn surface compensates the holes that are modulation-doped into the Sn reconstruction. STS data showing a ~ 0.2 eV upward shift of the surface chemical potential upon alkali adsorption confirm that alkali atoms donate electrons to the surface. STM imaging reveals that the area fraction of the $4\sqrt{3}\text{Sn}$ phase decreases from $\sim 40\%$ to nearly zero at 60 K with 0.005 ML potassium adsorbed on the $B-2\sqrt{3}\text{Sn}$ surface. This result, reversing the effects of modulation hole doping by on-surface electron doping, conclusively demonstrates that charge doping is indeed the root cause of the observed tip-assisted phase transformation on hole-doped samples. We present these results in Page 9 of the main text, Supplementary Note 10 and Supplementary Figure 12.

In conclusion, the manuscript presents promising results that can open new perspectives if finally confirmed. However, still these conclusions should be better experimentally justified before publication.

We thank Referee 1 for his or her constructive comments, which clearly have helped us to fully solidify our conclusions.

Responses to Referee 2 & 3

We thank both referees for their positive assessments on our previous revision and their recommendations for publication without change.

Reviewers' Comments:

Reviewer #1 (Remarks to the Author)

After checking the new figures and the comments added to the main text and the supplementary information I consider that now the paper is more solid, the authors have done an important effort to improve it. After all, I recommend the paper for the publication.